# Molecular Analysis of Polymyxin Resistance among Carbapenemase-Producing *Klebsiella pneumoniae* in Colombia

**DOI:** 10.3390/antibiotics10030284

**Published:** 2021-03-10

**Authors:** Elsa De La Cadena, María Fernanda Mojica, Juan Carlos García-Betancur, Tobías Manuel Appel, Jessica Porras, Christian José Pallares, Juan Sebastián Solano-Gutiérrez, Laura J. Rojas, María Virginia Villegas

**Affiliations:** 1Grupo de Investigación en Resistencia Antimicrobiana y Epidemiologia Hospitalaria, Universidad El Bosque, Bogotá 110121, Colombia; mfm72@case.edu (M.F.M.); betancurjuan@unbosque.edu.co (J.C.G.-B.); tappel@unbosque.edu.co (T.M.A.); jalporras@unbosque.edu.co (J.P.); cpallares@unbosque.edu.co (C.J.P.); mvvillegas@unbosque.edu.co (M.V.V.); 2Department of Infectious Diseases, Case Western Reserve University, Cleveland, OH 44106-7164, USA; ljr61@case.edu; 3Research Service, Louis Stokes Veterans Affairs Medical Center, Cleveland, OH 44106-7164, USA; 4Comité de Infecciones y Vigilancia Epidemiológica, Clínica Imbanaco, Cali 760031, Colombia; 5Departamento de Ciencias Biológicas, Escuela de Ciencias, Universidad EAFIT, Medellín 050022, Colombia; juanssolano2@gmail.com

**Keywords:** *Klebsiella pneumoniae*, multidrug resistance, polymyxins, whole-genome sequencing

## Abstract

Polymyxin resistance in *Klebsiella pneumoniae* has been attributed to mutations in *mgrB*, *phoPQ*, *pmrAB*, and *crrAB* and to the presence of *mcr* plasmid-mediated genes. Herein, we describe the molecular characteristics of 24 polymyxin- and carbapenem-resistant *K. pneumoniae* isolates recovered from six Colombian cities between 2009 and 2019. Minimum inhibitory concentrations (MICs) to polymyxin were confirmed by broth microdilution, and whole-genome sequencing was performed to determine sequence type, resistome, and mutations in the genes related to polymyxin resistance, as well the presence of *mcr*. The results showed high-level resistance to polymyxin (MICs ≥ 4 μg/mL). *bla*_KPC-3_ was present in the majority of isolates (17/24; 71%), followed by *bla*_KPC-2_ (6/24; 25%) and *bla*_NDM-1_ (1/24; 4%). Most isolates belonged to the CG258 (17/24; 71%) and presented amino acid substitutions in PmrB (22/24; 92%) and CrrB (15/24; 63%); mutations in *mgrB* occurred in only five isolates (21%). Additional mutations in *pmrA*, *crrA*, and *phoPQ* nor any of the mcr resistance genes were identified. In conclusion, we found clonal dissemination of polymyxin and carbapenem-resistant *K. pneumoniae* isolates in Colombia, mainly associated with CG258 and *bla*_KPC-3_. Surveillance of this multidrug-resistant clone is warranted due to the limited therapeutic options for the treatment of carbapenem-resistant *K. pneumoniae* infections.

## 1. Introduction

*Klebsiella pneumoniae* is a notorious nosocomial pathogen given its worldwide distribution and its capacity to develop resistance to multiple classes of antibiotics, including aminoglycosides, fluoroquinolones, extended-spectrum cephalosporins, and carbapenems [1]. Carbapenems are the preferred treatment for infections caused by *K. pneumoniae* isolates harboring extended-spectrum β-lactamases (ESBLs) [2,3]. Infections caused by carbapenem-resistant *K. pneumoniae* are difficult to treat and associated with high mortality rates, on average 41% [4]. Resistance to carbapenems has increased in *K. pneumoniae*, mainly due to the production of carbapenemases belonging to Ambler class A (KPC), class B (NDM and VIM), and class D (OXA-48). Polymyxins have re-emerged in clinical practice due to the limited antibiotic development pipeline and worldwide increasing prevalence of nosocomial infections caused by multidrug-resistant (MDR) Gram-negative bacteria including carbapenemase-producing *K. pneumoniae*. Polymyxin B and colistin (polymyxin E) have been ultimately considered as the last-resort treatment of such infections [5].

Polymyxins are cyclic polypeptides, with primary amines carrying a net polycationic charge at physiological pH and a hydrophobic fatty-acyl side chain. The amphipathic nature of polymyxins enables them to form ionic and hydrophobic molecular interactions at the bacterial outer membrane. Specifically, polymyxins initially bind to the lipid A of the lipopolysaccharide (LPS) through an ionic interaction that leads to the disruption of the outer membrane and the hydrophobic insertion of the fatty acyl side chain of polymyxin into lipid A. Subsequently, cytoplasmic membrane disruption causing cell death. Naturally, the increased use of polymyxins has led to the appearance of diverse mechanisms of polymyxin resistance, particularly during the course of therapy [6].

Resistance to polymyxin is provided by the addition of cationic groups to lipid A, either via chromosomally encoded pathways or plasmid-borne mobile colistin resistance gene (*mcr*) [7]. In *K. pneumoniae*, modification in several two-component systems such as PhoPQ, PmrAB, and CrrAB, as well as partial deletions or inactivation by different insertion sequence (IS) of the *mgrB* gene are currently the most commonly reported mechanisms of resistance [8]. In addition to these mechanisms, Jana et al. identified 35 nonessential chromosomal genes involved in polymyxin resistance among *K. pneumoniae* ST258 strains. Of those genes, collectively called “secondary resistome”, *dedA* encoding a putative integral membrane was shown to have a prominent role, as its deletion completely restored colistin susceptibility [9].

In Colombia, several molecular epidemiological studies have established that *K. pneumoniae*-producing KPC is endemic and represents a significant proportion of all clinical *K. pneumoniae* isolates in the country [10]. Furthermore, half of them belong to the clonal group (CG) 258 [11], which has been associated with the emergence of polymyxin resistance in *K. pneumoniae* in several countries [2,11,12]. Nevertheless, data of the prevalence of polymyxin-resistant *K. pneumoniae* in the country are scarce, mainly due to the shortcomings of the routine laboratory testing for polymyxin susceptibility [2,13]. Therefore, in this study we aimed to determine the profile and mechanism of polymyxin resistance in carbapenem-resistant *K. pneumoniae* isolates from six Colombian cities over a period of 10 years. Hereby, we describe the clonal dissemination of polymyxin and carbapenem-resistant *K. pneumoniae* isolates in several Colombian cities, mainly due to simultaneous mutations in *mgrB*, *pmrB*, and *CrrB* associated with the high-risk clone ST258.

## 2. Results

### 2.1. Antimicrobial Susceptibility

The susceptibility profile of the 24 isolates of *K. pneumoniae* included in this study is shown in Table 1. According to the standard definitions for acquired resistance proposed by Magiorakos et al. [14], all isolates exhibited multidrug-resistant phenotypes and in particular, were resistant to polymyxin with MICs ranging between 4 to ≥8 mg/L. The results also revealed high-level resistance to ceftazidime, cefepime, piperacillin-tazobactam, ertapenem, and ciprofloxacin. Ceftazidime-avibactam had the highest susceptibility percentage; as expected, only isolates producing NDM were resistant to it.

### 2.2. Molecular Typing and Virulome Analysis

According to the MLST analysis shown in Figure 1, the 24 isolates belonged to five different sequence types (STs). Most isolates belonged to CG258 (71%) (ST258, 16/24 and ST512, 1/24), followed by the ST14 (4/24, 17%), ST129 (2/24, 8%), and ST219 with one isolate (4%). Most of the KPC-3-*K. pneumoniae* producers belonged to the CG258 (16/17; 94%), while the isolate producing NDM-1 belonged to ST129 (Figure 1). Phylogenomic analysis performed using the CSI phylogeny tool grouped the different polymyxin-resistant *K. pneumoniae* isolates in two different clusters, coinciding with the presence of the *bla*_KPC-3_ and *bla*_KPC-2_ and in agreement with the MLST analysis, which clustered all CG258 isolates together (Figure 1). Regarding to virulence genes, KPC-3-producing *K. pneumoniae* belonging to ST258 and ST512 harbored multiple variants of distinct type 3 fimbriae genes, the yersiniabactin siderophore (*ybt*), and the genotoxic colibactin (*clb*), which have been associated with the type of capsular 154 (Appendix A). In addition, the analysis of plasmid types revealed sequences associated with the incompatibility groups colRNAI, IncFIB (K), IncFII (K), IncFII (YP), IncI2, and IncR (Appendix A).

### 2.3. Resistome Analysis

A wide variety of antibiotic resistance determinants to aminoglycosides, β-lactams, fluoroquinolones, macrolides, chloramphenicol, sulfonamide, fosfomycin, phenicol, tetracycline, co-trimoxazole, and rifampin were found within the genome of the isolates studied (Figure 1). Regarding quinolone resistance, analysis revealed the presence of several plasmid-mediated quinolone resistance (PMQR) genes such as *qnr*B, *qnr*S, and aac(6′)lb-cr. Additionally, mutations were found in the quinolone resistance-determining region (QRDR) of *gyrA*, leading to amino acid changes in Ser83Ile and Asp87Asn as well as in *parC*, leading to the amino acid change in Ser80Ile. The presence of oqxA and *fosA* was found in all isolates. Most of the isolates harbored *bla*_KPC_, being *bla*_KPC-3_ the most prevalent as it was detected in 17 of 24 isolates (71%); *bla*_KPC-2_ was carried by six isolates (25%), and only one isolate harbored *bla*_NDM-1_. On the other hand, analysis of the genetic context of the *bla*_KPC_ revealed that the majority of isolates carried this gene within a *Tn*4401b structure (87%), followed by *Tn*4401a (8.7%), and not-*Tn*4401 (4.3%) (Figure 1).

### 2.4. Mechanisms of Resistance towards Polymyxins

Amino acid sequences from other chromosomal genes found in these *K. pneumoniae* isolates associated with resistance to polymyxin were compared to those available in the GenBank, and the changes found are listed in Figure 2. Mutations in *ph*oPQ and *crrA* genes were not detected. We found changes in *mgrB* gene in five isolates (21%), including the presence of an IS-interrupting the *mgrB* gene in three of the 24 isolates (12.5%). Insertions of the IS element within the gene were found at nucleotides 74 and 99. Moreover, in two isolates (8%), *mgr*B was prematurely terminated due to a single-nucleotide change at position 88 causing the introduction of a stop codon, which generates a shorter protein of only 29 amino acids (Figure 2).

On the other hand, sequence analysis of the *pmrB* genes revealed several non-synonymous mutations that yield the following amino acid substitutions in 22 polyxymin resistant isolates: T157P, N105K, R256G, and T246A (Figure 2). According to the prediction of the SMART tool (http://smart.embl-heidelberg.de/) (accessed on 28 July 2020), T157 is located in the histidine kinase A (HisKA) phosphoacceptor domain, N105 and R256 in the HATPase_c domain, and T246 does not appear to fall within any conserved domain. In contrast, one-point mutation in the *pmrA* was found in only one isolate. In regard to CrrB, we identified three amino acid substitutions in this protein (C68S, G183V, and Q296L) among 15 of the 24 polymyxin-resistant isolates examined. Structurally, C68 is in the transmembrane region, G183 in the histidine kinase A (HisKA) phosphoacceptor domain and Q296 in the HATPase_c domain. The *crrB* gene was absent from nine isolates (Figure 2). Interestingly, *mcr* genes were not detected in any of the isolates.

## 3. Discussion

Several studies have shown that mutations identified within the *mgrB* gene and its insertional inactivation mediated by mobile IS are the most prevalent mechanisms involved in the development of polymyxin resistance in *K. pneumoniae* [15]. Our results confirmed these findings and showed that polymyxin resistance was associated to loss of function in *mgrB* and mutations in the *pmrB* and *crrB* genes. Major loss of function mutations in *mgrB* (premature stop codon, insertional inactivation, or deletion) were detected in five (21%) of the polymyxin-resistant strains (Figure 2). In three isolates (12.5%), the inactivation of *mgrB* gene was identified by insertion of IsKpn26 and two with premature stop codons. Jaidane et al. showed that colistin resistance in *K. pneumoniae* was directlylinked to mutations identified within the MgrB protein [16]; however, we did not find any amino acid substitutions in MgrB. Other studies have shown that the insertion of IS5 element into the *mgrB* gene was involved in resistance to polymyxins [15,17]. A large number of different ISs have been associated with transposition into *mgrB* including IS5, ISL3, IS5-like, IsKpn13, ISKpn14, and ISKpn26 with the IS5 family being the most commonly reported [8,18,19]. Surprisingly, in the isolates analyzed in this study, an IS5-like sequence was inserted at the same location in two of the isolates (UEB_12 and UEB_13). Insertional inactivation of *mgrB* occurred at nucleotide 74 and was caused by an IS that shares identity with ISKpn26 and belongs to the IS5 family [12,15]. The repeated observation of this insertional inactivation could indicate that is a hot spot for insertions. Likewise, Novovic et al. described one isolate with premature stop codon (TAG) due to a C-to-T change at position 88, which generates a truncated MgrB protein of 29 amino [20], just like the one found in the isolates UEB_14 and UEB_18. Truncations identified at positions 26, 28, and 30 of *mgrB* have also been previously described [21,22].

Sequence analysis of *pmrB* and *crrB* genes known to be involved in LPS synthesis revealed several polymorphisms in nucleotide sequences compared to the wild-type sequence of *K. pneumoniae* [18,23]. Notably, these polymorphisms were linked to specific STs. Within strains belonging to ST258, amino acid changes were observed in T157P, T246A, and R256G in PmrB, and C68S and Q296L in CrrB. Most belonging to the CG258 carried the double PmrB variant T246A R256G; in addition to these, the isolate identified as UEB_2 had also the T157P substitution. Interestingly, Jaidane et al. reported the T246A PmrB variant in *K. pneumoniae* ST111 isolates and the T246A R256G double PmrB variant in isolates belonging to other STs (392 and 247) collected in Tunisia [16]. Likewise, Jayol et al. described the PmrB variant T157P, which contributes to polymyxin resistance, in isolates recovered from different geographic regions and belonging to different STs. However, the Colombian isolate reported in that study carrying that mutation belonged to the CG258 [24]. On the other hand, the N105K substitution in PmrB was exclusively found among isolates belonging to the ST14. A different substitution in the same position (N105S) had been previously identified in K. pneumoniae isolates from Taiwan and South Korea [25], although molecular evidence about its role in conferring polymyxin resistance is not yet available. On the other hand, mutations in the *crrB* gene have also been associated with polymyxin resistance. Specifically, those leading to the Q10L, L94M, and N141Y amino acid substitutions have been shown to confer high-level resistance to polymyxin [18,26]. None of these *crrB* mutations were found in our analyzed isolates, and more studies are needed to elucidate the role of the C68S, G183V, and Q296L substitutions in the function of CrrB and their association with polymyxin resistance.

Different studies carried out in France with Colombian isolates mostly including ST258 isolates carrying *bla*_KPC-2_ and *bla*_KPC-3_ found mutations in CrrB (G183V), particularly in isolates producing KPC-3 [18]. Additionally, five isolates from the CG258 producing KPC-2 showed the insertion of different IS within the *mgrB* gene (IS5-like, IsKpn13, IsKpn14, Is10R) [12], and three isolates, two of them belonging to ST258 and producing KPC-3, presented a mutation in *pmrB* (T157P) [24]. In line with the previously reported molecular epidemiology of KPC-producing *K. pneumoniae* isolates in Colombia [10,27,28], 17 of the 24 isolates (71%) belonged to the epidemic CG258, and 16/17 (94%) harbored the *bla*_KPC-3_. As shown in Figure 1, WGS phylogenomic analysis effectively clustered the polymyxin-resistant *K. pneumoniae* isolates in agreement with both the presence of the *bla*_KPC-3_ and *bla*_KPC-2_ and the clustering obtained by MLST analysis, demonstrating a clonal pattern. Consistently with the clonal spread reported in this study, large outbreaks of polymyxin-resistant, KPC-3-harboring *K. pneumoniae* from ST258 and ST512 (SLV of ST258) have been reported in Israel, Italy, Greece, and the United States [10,29,30].

Interestingly, isolates found within the two main clusters have diverse geographical origins and additionally were isolated in a broad time range (from 2009 to 2019). Nevertheless, the close genetic relatedness of isolates collected from the same geographical region that belong to the same ST group and carry the same *bla*_KPC_ gene, suggests that they also tend to share a particular polymyxin-resistance profile. Indeed, the inactivation of *mgrB* by ISKpn26 was only observed in isolates from one hospital in Medellin (M2) belonging to the ST258 (Figure 2). Cienfuegos-Gallet et al. reported similar results in *K. pneumoniae* isolates from this Colombian city [31]. In that study, they analyzed 32 carbapenem and colistin-resistant *K. pneumoniae* isolates, of which 27 belonged to ST512. They reported the presence of ISKpn25 within the *mgrB* gene in a cluster of 19 isolates from the same clonal group isolated from the same hospital. Likewise, additional mutations in *phoPQ*, *pmrAB*, and *crrAB* were not identified [31].

Of note, in accordance with the secondary resistome associated with colistin resistance in *K. pneumoniae* reported by Jana et al., [9] we found that all our polymyxin resistant isolates carried wild type *dedA*. Lastly, *mcr* genes were not detected suggesting that although this gene has been found in clinical *Enterobacterales* isolates, including *K. pneumoniae* [32], its prevalence among this species remains low. Therefore, resistance to polymyxin in isolates of *K. pneumoniae* in Colombia appears to be mainly restricted to chromosomal mutations, as previously described by Berglund et al. [33].

## 4. Materials and Methods 

### 4.1. Bacterial Isolates and Antimicrobial Susceptibility

A total of 24 nonduplicate polymyxin and carbapenem resistant *K. pneumoniae* isolates from nine hospitals from six different Colombian cities were used in this study. These clinical isolates were collected through the Colombian Bacterial Resistance Surveillance Network between 2009 and 2019. Species identification was performed using the VITEK2 system (bioMerieux, Marcy l’Etoile, France). The minimum inhibitory concentration (MIC) of ceftazidime, cefepime, piperacillin-tazobactam, ertapenem, imipenem, meropenem, doripenem, ceftazidime-avibactam, ciprofloxacin, and tigecycline was determined by broth microdilution assay using customized Sensititre plates (Sensititre™; Thermo Fisher, Waltham, MA, USA) and interpreted according to the Clinical and Laboratory Standards Institute guidelines (CLSI) [34]. Resistance was determined according to CLSI guidelines except for tigecycline for which the European Committee on Antimicrobial Susceptibility Testing (EUCAST) was used (http://www.eucast.org) (accessed on 16 February 2020). Polymyxin resistance was interpreted according to CLSI (MIC > 2 mg/L), based on the macrodilution method with polymyxin sulfate (Sigma-Aldrich, St. Louis, MO, USA) and cation-adjusted Mueller-Hinton broth according to CLSI M7 guidelines [35]. *Escherichia coli* ATCC 25922 (polymyxin susceptible) and an *E. coli* isolate 6770 harboring the mcr-1 gene previously characterized (GenBank: MVPG00000000.1) [32] and with an expected MIC > 4 g/mL) were used as control strains for antimicrobial susceptibility testing (Table 1).

### 4.2. Molecular Typing

The presence of carbapenemases was confirmed by qPCR as published by Correa et al. [36]. A total of 24 polymyxin-resistant and carbapenemase-producing *K. pneumoniae* isolates were whole-genome sequenced (WGS) using the Illumina Nextera XT library prep kit and the Illumina MiSeq sequencing platform (Illumina, San Diego, CA, USA). Sequence annotation was done in PATRIC, the bacterial bioinformatics database and analysis resource (http://www.patricbrc.org) (accessed on 20 February 2020) [37]. Genome assemblies were uploaded to the Center for Genomic Epidemiology (CGE; https://cge.cbs.dtu.dk) (accessed on 25 February 2020) to identify antibiotic-resistance genes, plasmidic profiles, and STs, using the ResFinder, Plasmid Finder database [38,39], and the MLST (multilocus sequence typing) 1.8 server, respectively (the MLST scheme included: *rpoB, gapA, mdh, pgi, phoE, infB*, and *tonB*) [40]. Virulence genes and capsular typing via the *wzi* gene were performed in silico using the typing scheme at BIGSdb (http://bigsdb.pasteur.fr/) (accessed on 9 March 2020) [41]. To identify the genetic environment of the *bla*_KPC_ gene, sequences were mapped into the *Tn*4401 transposon sequence (GenBank accession number KT378597.1), which was used as reference. The consensus sequences from the mapping were compared with *tnp*R, *tnp*A, *ist*B, *ist*A, and *bla*K_PC_ by BLAST analysis.

Insertion sequences (IS) and mutations in *mgrB* gene or in the two-component regulatory systems *pmrAB*, *phoPQ*, and *crrAB* were searched within these isolates. The insertional sequence was identified using the ISfinder database (http://www-is.biotoul.fr) (accessed on 12 May 2020) [42], and the resulting sequences were compared to those of the polymyxin-susceptible strain *K. pneumoniae* (GenBank accession number CP000647). Whole-genome, single-nucleotide-polymorphism (SNP) phylogeny among isolates was inferred using the CSI Phylogeny version 1.4 (https://cge.cbs.dtu.dk/services/CSIPhylogeny/) (accessed on 21 March 2020) [43] to determine the epidemiological relationship between the isolates. To infer phylogenies using this SNP alignment, we used all the default values suggested by the developers, including the FastTree method (PMID 19377059), which infers phylogenies from big sequence-datasets based on the ‘‘minimum-evolution’’ principle instead of distance matrixes and uses significantly lower computational times. FastTree provides local support values based on the Shimodaira Hasegawa (SH) test (DOI: https://doi.org/10.1093/oxfordjournals.molbev.a026201) (accessed on 10 September 2020) with 1,000 bootstrap replicates to estimate the confidence in the given split. The graphical representation of the resulting phylogeny was obtained using the iTol interactive online tool (https://itol.embl.de/) (accessed on 21 September 2020) [44]. Variants were called in relation to the reference genome of the *K. pneumoniae* strain 30660/NJST258_1 (NCBI accession CP006923.1). Sequencing data of the polymyxin-resistant and carbapenemase-producing *K. pneumoniae* isolates from this study are available in NCBI BioProjects: PRJNA657895.

## 5. Conclusions

Herein, we report that the endemic CG258 represented by ST258 and ST512 (17 of 24 strains) exhibit high-level resistance to polymyxin. Since isolates belonging to this clonal group are one of the main drivers of the expansion of *bla*_KPC_ in the country, the massive use of polymyxins to treat infected patients with carbapenemase-producing *K. pneumoniae* may lead to selective pressure and increase resistance. Active surveillance is warranted to prevent further spread of this clone in the country and to track the appearance of more polymyxin resistance determinants, including *mcr-1*.

## Figures and Tables

**Figure 1 antibiotics-10-00284-f001:**
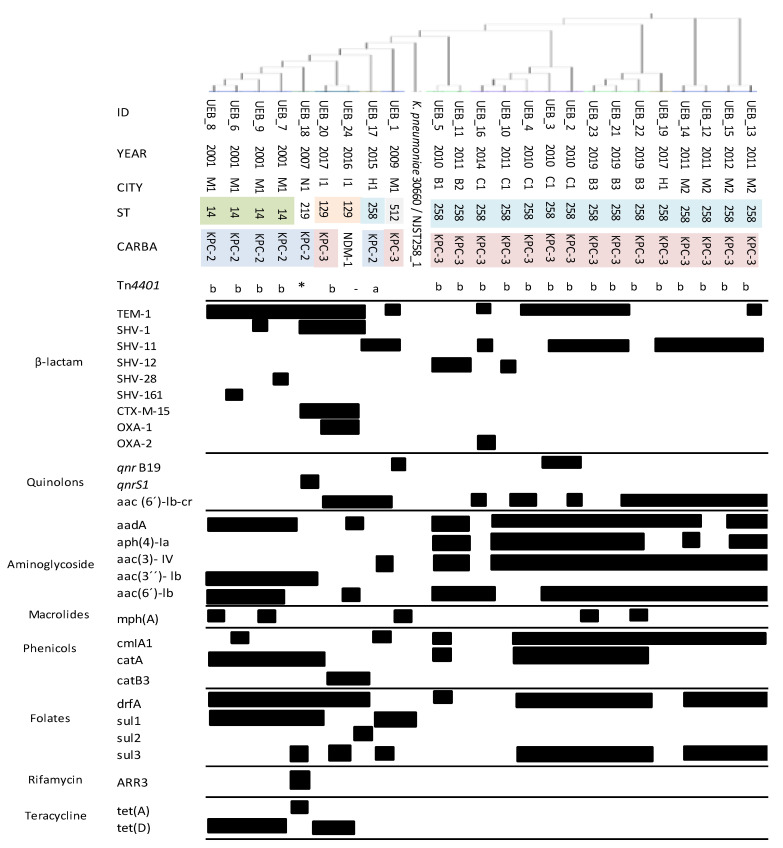
Genomic characteristics of the 24 *K. pneumoniae* isolates included in this study. C1: Cali, M1: institution 1 Medellin, M2: institution 2 Medellin, B1: Institution 1 Bogotá, B2: institution 2 Bogotá, B3: institution 3 Bogotá, N1: Neiva, I1: Ibague, H1: Pasto. Black indicates presence, * not-*Tn4401.*

**Figure 2 antibiotics-10-00284-f002:**
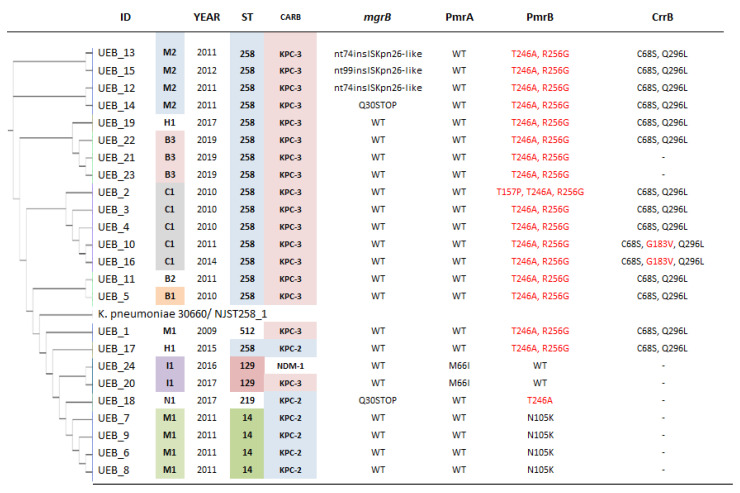
Characteristics of polymyxin resistant *K. pneumoniae* isolates included in this study. C1: Cali, M1: institution 1 Medellin, M2: institution 2 Medellin, B1: Institution 1 Bogotá, B2: institution 2 Bogotá, B3: institution 3 Bogotá, N1: Neiva, I1: Ibague, H1: Pasto. Mutations previously reported associated with polymyxin resistance are highlighted in red. Q30STOP: Premature stop codon at 30th amino acid. The interactive tree of life (iTOL) tool was used for the phylogenetic tree display and annotation. Bootstrap values shown in Appendix A.

**Table 1 antibiotics-10-00284-t001:** Characteristics and minimum inhibitory concentrations of polymyxin-resistant *K. pneumoniae* (mg/L).

ID	HOSPITAL	YEAR	SOURCE	ROOM	POL	CZA	CAZ	FEP	TZP	ETP	IMI	MEM	DOR	CIP	TGC
UEB_1	M1	2009	BLOOD	HOSP	4	4/4	>32	>64	>64/4	>32	>32	>32	>16	>2	4
UEB_2	C1	2010	SKIN	ER	8	≤1/4	>32	>64	>64/4	>32	32	>32	>16	>2	≤0.5
UEB_3	C1	2010	BLOOD	ICU	8	4/4	>32	>64	>128/4	>32	32	>32	>16	>2	≤0.5
UEB_4	C1	2010	SKIN	ICU	>8	4/4	>32	>64	>128/4	>32	>32	>32	>16	>2	1
UEB_5	B1	2010	OTHER	HOSP	4	≤1/4	>32	64	>128/4	2	1	1	1	>2	≤0.5
UEB_6	M1	2011	GI	ICU	>8	2/4	>32	16	>128/4	16	4	4	4	1	≤0.5
UEB_7	M1	2011	SECRETION	ICU	8	4/4	>32	16	>128/4	>32	16	>32	>16	1	1
UEB_8	M1	2011	GI	ICU	>8	2/4	>32	>64	>128/4	>32	16	32	>16	1	1
UEB_9	M1	2011	GI	ICU	8	≤1/4	>32	16	128/4	2	1	1	≤0.5	1	1
UEB_10	C1	2011	GI	ICU	>8	≤1/4	>32	64	>128/4	32	4	16	16	>2	2
UEB_11	B2	2011	BLOOD	ICU	>8	4/4	>33	>64	>128/4	>32	>32	>32	>16	>2	2
UEB_12	M2	2011	RESP TRACT	ICU	>8	4/4	>32	>64	>128/4	>32	>32	>32	>16	>2	1
UEB_13	M2	2011	BLOOD	ICU	>8	4/4	>32	64	>128/4	>32	>32	>32	>16	>2	≤0.5
UEB_14	M2	2011	GI	ICU	>8	2/4	>32	64	>128/4	>32	>32	>32	>16	>2	1
UEB_15	M2	2012	URINE	HOSP	>8	4/4	>32	64	>128/4	>32	>32	>32	>16	>2	1
UEB_16	C1	2014	BLOOD	HOSP	>8	≤1/4	>32	>64	>128/4	32	>32	>32	>16	>2	4
UEB_17	H1	2015	RESP TRACT	ICU	>8	2/4	>32	64	>128/4	>32	>32	>32	>16	1	2
UEB_18	N1	2017	BLOOD	ER	>8	≤1/4	>32	32	>128/4	4	2	1	2	≤0.25	4
UEB_19	H1	2017	URINE	HOSP	>8	≤1/4	>32	64	>128/4	>32	32	32	>16	>2	4
UEB_20	I1	2017	URINE	HOSP	>8	≤1/4	>32	>64	>128/4	32	8	16	16	1	2
UEB_21	B3	2019	RESP TRACT	HOSP	>8	4/4	>32	>64	>128/4	>32	>32	>32	>16	>2	1
UEB_22	B3	2019	PERIT LIQ	HOSP	>8	2/4	>32	>64	>128/4	>32	>32	>32	>16	>2	1
UEB_23	B3	2019	ASCIT LIQ	HOSP	>8	4/4	>32	>64	>128/4	>32	>32	>32	>16	>2	1
UEB_24	I1	2016	RESP TRACT	HOSP	>8	>64/4	>32	>64	>128/4	>32	>32	32	>16	>2	2

POL: polymyxin, CZA: ceftazidime-avibactam, CAZ: ceftazidime, FEP: cefepime, TZP: piperacillin-tazobactam, ETP: ertapenem, IMI: imipenem, MEM: meropenem, DOR: doripenem, CIP: ciprofloxacin, and TGC: tygecicline. C1: Cali, M1: institution 1 Medellin, M2: institution 2 Medellin, B1: Institution 1 Bogotá, B2: institution 2 Bogotá, B3: institution 3 Bogotá, N1: Neiva, I1: Ibague, H1: Pasto. ICU: intensive care unit, ER: emergency room, and HOSP: hospitalization room. GI: gastrointestinal, SKIN: Skin and Soft Tissue, RESP TRACT: respiratory Tract, PERIT LIQU: peritoneal liquid, ASCITIC LIQ: ascitic liquid.

## Data Availability

Sequencing data of the polymyxin-resistant and carbapenemase-producing *K. pneumoniae* isolates from this study are available in NCBI BioProject: PRJNA657895.

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
