# Peer review of "Molecular Analysis of Polymyxin Resistance among Carbapenemase-Producing Klebsiella pneumoniae in Colombia"

_antibiotics, 2021, doi:10.3390/antibiotics10030284_

Round 1

Reviewer 1 Report

          Antibiotic resistance is a global problem. The manuscript by Cadena et al analyzed the Polymyxin resistance among Carbapenemase-producing Klebsiella pneumoniae in Colombia. Using molecular techniques such as MIC by broth dilution and whole-genome sequencing were used to know the sequence type, resistome, and mutations in the genes related to polymyxin resistance. They found that the spreading of polymyxin and carbapenem-resistant K. pneumoniae isolates in numerous Colombian cities related to simultaneous mutations in mgrB, pmrB, and CrrB related to the high-risk clone ST258.

          Although not novel (many similar studies have been done in a different part of the world) this study reported additional mutations in genes that play a role in polymyxin resistance. Overall the study is well done, the manuscript is clearly written and conclusions are justified. In my opinion, the manuscript is likely to be of interest to a diverse readership, including those interested in the antibiotic resistance mechanism.

There are a few suggestions/questions to authors which in my opinion will further improve the content of the manuscript:

Minor issues:

  1. Here and there in the manuscript, there are some typos that need further proofreading.
  2. In my opinion, line 62-66, the author should consider discussing the role of secondary resistome in the colistin and polymyxin resistance (PMID: 28198411)(PMID: 31827463).
  3. While conducting the whole genome analysis on the isolates does the author encountered any of the secondary resistome genes as reported in the (PMID: 28198411)? It will be interesting to know.

Reviewer 2 Report

The work presented by De La Cadena et al. studies the polymyxin resistance among carbapenemase producing Klebsiella pneumoniae in Colombia. To this end, whole-genome sequencing and other molecular analyses of 24 isolates from different Colombian cities were performed. The results revealed the presence of different mutations associated to antibiotic resistance.

The study overall is very interesting and well presented. I recommend accepting the manuscript after the following revisions.

Specific comments

Line 23: Please, replace twenty-four by 24.

Line 46: "average"

Line 68: It says several studies but only one is cited. Please, clarify this.

Line 80: Please, place the Materials and Methods section before the Results section.

Line 91: What does ST mean? Please, clarify this.

Lines 142-144: Are those isolates denoted as Q30STOP in Figure 2? Please, clarify this point.

Lines 153-154: Please, place the reference to pmrA in a different sentence. Also, please check the next sentence as it lacks a full stop.

Line 161: Please, provide bootstrap or other branch support value to the phylogenetic tree.

Line 170: Please, replace 2 by "two".

Line 177: IS-5 is the same as ISKpn26? The figure 5 shows the ISKpn26-like. Also it mentions nt74 and not 75 as discussed later.

Line 177: which isolates?

Line 202: Please, replace 5 by five.

Line 203: Please, replace 3 by three.

Line 232. As mentioned earlier, please, place the Materials and Methods section before the Results section.

Line 235: Is this (9) a citation? Please, clarify this point.

Line 236: Please, replace 6 by six.

Line 254: Please provide more details on how the qPCR was performed (i.e. primers, PCR conditions, etc.) and the method employed for DNA extraction.

Line 261: Which genes or sequences were analyzed with MLST?

Line 278: Was this accession released? It could not be accessed from NCBI. Pease, check it.

Line 310: Please, check that the scientific names are written properly in the References section (i.e. italics).
